# A Non-parametric Learning Method for Confidently Estimating Patient's Clinical State and Dynamics

**William Hoiles**
Department of Electrical Engineering
University of California Los Angeles
Los Angeles, CA 90024
whoiles@ucla.edu

**Mihaela van der Schaar**
Department of Electrical Engineering
University of California Los Angeles
Los Angeles, CA 90024
mihaela@ee.ucla.edu

## Abstract

Estimating patient's clinical state from multiple concurrent physiological streams plays an important role in determining if a therapeutic intervention is necessary and for triaging patients in the hospital. In this paper we construct a non-parametric learning algorithm to estimate the clinical state of a patient. The algorithm addresses several known challenges with clinical state estimation such as eliminating the bias introduced by therapeutic intervention censoring, increasing the timeliness of state estimation while ensuring a sufficient accuracy, and the ability to detect anomalous clinical states. These benefits are obtained by combining the tools of non-parametric Bayesian inference, permutation testing, and generalizations of the empirical Bernstein inequality. The algorithm is validated using real-world data from a cancer ward in a large academic hospital.

## 1   Introduction

Timely clinical state estimation can significantly improve the quality of care for patient's by informing clinicians of patient's that have entered a high-risk clinical state. This is a challenging problem as the patient's clinical state is not directly observable and must be inferred from the patient's vital signs and the clinician's domain-knowledge. Several methods exist for estimating the patient's clinical state including clinical guidelines and risk scores [21, 18]. The limitation with these population based methods is that they are not personalized (e.g. patient models are not unique), can not detect anomalous patient dynamics, and most importantly, are biased due to therapeutic intervention censoring [16]. Therapeutic intervention censoring occurs when a patient's physiological signals are misclassified in the training data as a result of the effects caused by therapeutic interventions. To improve the quality of patient care, new methods are needed to overcome these limitations.

In this paper we develop an algorithm for estimating a patient's clinical state based on previously recorded electronic health record (EHR) data. A schematic of the algorithm is provided in Fig.1 which contains three primary components: a) learning the patient's stochastic model, b) using statistical techniques to evaluate the quality of the estimated stochastic model, and c) performing clinical state estimation for new patients based on their estimated models. The works by Fox et al. [10, 9] and Saria et al. [19] for temporal segmentation are the most related to our algorithm. However [10, 19] do not apply formal statistical techniques to validate and iteratively update the hyper-parameters of the non-parametric Bayesian inference, are not personalized, do not remove the bias caused by therapeutic intervention censoring, and do not utilize clinician domain knowledge for clinical state estimation. Additionally, applying fully Bayesian methods [9] for clinical state estimation are computationally prohibitive as the computational complexity of constructing the stochastic model of all patients grows polynomially with the number of samples and maximum number of possible states of all patients. The computational complexity of our algorithm is only polynomial in the number

of samples and states of a single patient. A detailed literature review is provided in the Supporting Material.

The proposed algorithm (Fig.1) learns a combinatorial stochastic model for each patient based on their measured vital signs. A non-parametric Bayesian learning algorithm based on the hierarchical Dirichlet process hidden Markov model (HDP-HMM) [10] is used to learn the patient's stochastic model which is composed of a possibly infinite state-space HMM where each state is associated with a unique dynamic model. The algorithm dynamically adjusts the number of detected dynamic models and their temporal duration based on the patient's vital signs–that is, the algorithm has a data-driven bound on the model complexity (e.g. number of detected states). The patient's stochastic model provides a fine-grained personalized representation of each patient that is interpretable for clinicians, and accounts for the patient's specific dynamics which may result from therapeutic interventions and medical complications (e.g. disease, paradoxical reaction to a drug, bone fracture). To ensure that each detected dynamic model is associated with a unique clinical state, the hyper-parameters in the HPD-HMM are updated iteratively using the results from an improved Bonferroni method [2]. This mitigates the major weakness of non-parametric Bayesian inference methods of how to select the hyper-parameters [14, 12]. Additionally, the algorithm provides statistical guarantees on the dynamic model parameters using generalizations of the scalar Bernstein inequality [13] to vector-valued and matrix-valued random variables. In clinical applications it is desirable to relate a collection of dynamic models from several patient's to a unique clinical state of interest for the clinician (e.g. detecting which patients have entered a high-risk clinical state). The clinician defines a supervised training set that is composed of all previously observed patient's dynamic models and their associated clinical state, which is then used to construct a similarity metric. This construction of the similarity metric between dynamic models and clinical states ensures that the bias introduced from therapeutic intervention censoring is removed, and also allows for the detection of anomalous dynamic models that are not associated with a previously defined clinical state. When a new patient arrives the algorithm will learn their stochastic model, and then use the similarity metric to map the detected dynamic models to their associated clinical states of interest.

Though our algorithm is general and can be applied in several medical settings (e.g. mobile health, wireless health) here we focus on detecting the clinical state of patients in hospital wards. Specifically we apply our algorithm to patient's in a cancer ward of a large academic hospital.

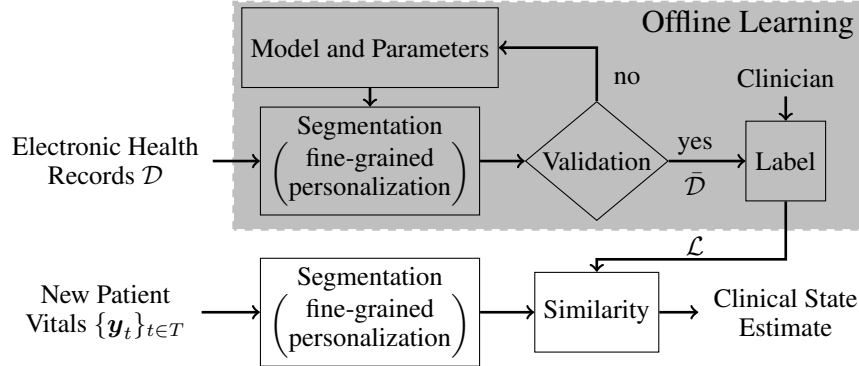

Figure 1: Schematic of the proposed algorithm for learning the dynamic model and estimating the clinical state of the patient. From $\mathcal{D}$ a valid segmentation $\bar{\mathcal{D}}$ is constructed and provided to the clinician to construct the labeled dataset $\mathcal{L}$. New patient vital signs are labeled using the dataset $\mathcal{L}$.

## 2 Non-parametric Learning Algorithm for Patient's Stochastic Model

In this section we provide a method to segment patient's electronic health record data $\mathcal{D} = \{\{y_t^i\}_{t\in T^i}\}_{i\in I}$, with $y_t^i \in \mathbb{R}^m$ the vital signs of patient $i \in I$ at time $t$. To segment the temporal data we assume that the vital signs of each patient originate from a switching multivariate Gaussian (SMG) process. A Bayesian non-parametric learning algorithm is utilized to select the switching times between the unique dynamic models–that is, we consider the observation dynamics and model switching dynamics simultaneously. The final result of the segmentation is the dataset:

$$\bar{\mathcal{D}} = \{\{y_t^i\}_{t\in T_k^i}, k \in \{1,\ldots,K^i\} = \mathcal{K}^i\}_{i\in I} \tag{1}$$

with $T_k^i$ the time samples for segment $k$ and $\mathcal{K}^i$ the set of segments for patient $i$. Statistical methods are used to ensure that each dynamic model is associated with a unique clinical state, refer to Sec.3 for details.

We assume that the switching process between models satisfies a HMM where each state of the HMM is associated with a unique dynamic model given by:

$$\boldsymbol{y}_t = \varepsilon_t(z_t) \qquad \varepsilon_t(z_t) \sim \mathcal{N}(\boldsymbol{\mu}(z_t), \Sigma(z_t)) \tag{2}$$

where $z_t \in \mathcal{K}^i$ is the state of the patient, and $\varepsilon_t(z_t)$ is a Gaussian white noise term with covariance matrix $\Sigma_{(z_t)}$. For notational convenience we will suppress the indices $i$ and only include explicitly when required. For segmentation each of the patients is treated independently. Each state $z_t$ is assumed to evolve according to a HMM with $z_t$ associated with a specific segment $k \in \mathcal{K}$. Notice that we must estimate the total number of states $|\mathcal{K}|$, and the associated model parameters $\{\mu(k), \Sigma(k)\}_{k \in \mathcal{K}}$ using only the data $\{y_t\}_{t \in T}$.

To learn the cardinality of the HMM we use the tools of non-parametric Bayesian inference by placing a prior on the HMM parameters to allow a data-driven estimation of cardinality of the state-space. Recall that non-parametric here indicates that for larger sample size $T$, the number of possible states (i.e. dynamic models) can also increase. To model the infinite-HMM we use the *hierarchical Dirichlet process* (HDP) [3, 22]. The HDP can be interpreted as a HMM with a countably infinite state-space. That is, the HDP is a non-parametric prior for the infinite-HMM. The main idea of the HDP is to link a countably infinite set of Dirichlet processes by sharing atoms among the DPs with each DP associated with a specific state. The stick-breaking construction of the HDP is given by [8, 22]:

$$m \sim H, \quad \phi_0 = \sum_{m=1}^{\infty} \beta_m \delta_m, \quad \beta_m = v_m \prod_{l=1}^{m-1}(1 - v_l), \quad v_m \sim \mathrm{Beta}(1, \gamma),$$

$$\phi_k = \sum_{m=1}^{\infty} \pi_{km} \delta_m, \quad \boldsymbol{\pi}_k \sim \mathrm{DP}(\alpha, \boldsymbol{\beta}). \tag{3}$$

Eq.(3) represents an infinite state HMM with $\pi_{km}$ the transition probability of transitioning from state $k \in \mathcal{K}$ to state $m \in \mathcal{K}$. $\boldsymbol{\pi}_k$ represents the transition probabilities out of state $k$ of the HMM with $\boldsymbol{\beta}$ the shared prior parameter of the transition distribution, $H$ is a prior on the transition probability distribution, and $\alpha$ the concentration of the transition probability distribution of the HMM.

The patient's stochastic model is constructed by combining the SMG (2) with the HDP (or infinite HMM) and is given by:

$$v_k \sim \mathrm{Beta}(1, \gamma), \quad \beta_k = v_k \prod_{l=1}^{k-1}(1 - v_l), \quad \boldsymbol{\pi_k} \sim \mathrm{DP}\left(\alpha + \kappa, \frac{\alpha \boldsymbol{\beta} + \kappa \delta_k}{\alpha + \kappa}\right) \quad k = 1, 2, \ldots$$

$$z_t \sim \pi(\cdot | z_{t-1}) = \boldsymbol{\pi}_{z_{t-1}}, \quad \boldsymbol{y}_t = \varepsilon(z_t) \quad t = 1, 2, \ldots, T. \tag{4}$$

The parameter $\gamma$ controls how concentrated the state transition function is from state $k$ to state $k'$. This can be seen by setting $\kappa = 0$ and $\alpha = 0$ such that $E[\boldsymbol{\pi}_k] = \boldsymbol{\beta}$. If $\gamma = 1$ then the parameter $\beta_k$ in $\boldsymbol{\beta}$ decays at approximately a geometric rate for increasing $k$. As $\gamma$ increases, the decay of the elements in $\boldsymbol{\beta}$ decrease. For $\alpha > 0$ and $\kappa > 0$ then $E[\boldsymbol{\pi}_k] = (\alpha \boldsymbol{\beta} + \kappa \delta_k)/(\alpha + \kappa)$, as such $\kappa$ controls the bias of $\boldsymbol{\pi}_k$ towards self-transitions–that is, $\pi(k|k)$ is given a large weight. The parameter $\alpha + \kappa$ controls the variability of $\boldsymbol{\pi}_k$ and the base state transition distribution $(\alpha \boldsymbol{\beta} + \kappa \delta_k)/(\alpha + \kappa)$.

Given the patient's stochastic model (4), non-parametric Bayesian inference are utilized to estimate the model parameters from the patient's vital signs $\{\boldsymbol{y}_t\}_{t \in T}$. To utilize Bayesian inference we define a prior and compute the associated posterior since a $\sigma$-finite density measure is present. The prior distributions on $\boldsymbol{\beta}$ and $\boldsymbol{\pi}$ are given by:

$$\boldsymbol{\beta} \sim \mathrm{Dir}(\gamma/L, \ldots, \gamma/L), \quad \boldsymbol{\pi}_k \sim \mathrm{Dir}(\alpha \beta_1, \ldots, \alpha \beta_k + \kappa, \ldots, \beta_L) \quad k \in \{1, \ldots, L\}. \tag{5}$$

Eq.(5) is the weak limit approximation with truncation level $L$ where $L$ is the largest number of expected states in the estimated HMM from $\{\boldsymbol{y}_t\}_{t \in T}$ [25]. Note that as $L \to \infty$ then (5) approach the HDP. If clinician domain knowledge is not available on the initial hyper-parameters $\gamma, \alpha$, and $\kappa$, then it is common to place Beta or Gamma priors on these distributions [25]. For the multivariate Gaussian we utilize the Normal-Inverse-Wishart prior distribution [11]:

$$p(\boldsymbol{\mu}, \Sigma | \boldsymbol{\mu}_0, \lambda, S_0, v) \propto |\Sigma|^{\frac{v+m+1}{2}} \exp\left(-\frac{1}{2} \mathrm{tr}(v S_0 \Sigma^{-1} - \frac{\lambda}{2}(\boldsymbol{\mu} - \boldsymbol{\mu}_0)' \Sigma^{-1}(\boldsymbol{\mu} - \boldsymbol{\mu}_0))\right) \tag{6}$$

where $v$ and $S_0$ are the degrees of freedom and the scale matrix for the inverse-Wishart distribution on $\Sigma$, $\boldsymbol{\mu}_0$ is the prior mean, and $\lambda$ is the number of prior measurements on the $\Sigma$ scale. Given the prior distribution with associated posterior distributions a MCMC or variational sampler (i.e. Gibbs sampler [10], Beam sampler [25], variational Bayes [6, 7]) can be utilized to estimate the parameters of the patient's stochastic model (4) given the data $\{\boldsymbol{y}_t\}_{t \in T}$.

## 3 Statistical Methods to Evaluate Stochastic Model Quality

Given the segmented dataset $\bar{\mathcal{D}}$ (1) generated from all the patient's estimated stochastic models (4), this section presents methods to evaluate the quality of $\bar{\mathcal{D}}$. This includes testing if the vital signs $\{y_t^i\}_{t \in T_k^i}$ for each patient and unique dynamic model are consistent with a multivariate Gaussian distribution, contain sufficient samples to guarantee the accuracy of the dynamic model parameters, and that the detected dynamic models for each patient are unique. If the estimated stochastic models are of low quality then the hyper-parameters of the non-parametric Bayesian inference algorithm can be iteratively updated to ensure that all the patient's stochastic models accurately represent their dynamics. This is a vital step in medical applications since the results of the non-parametric Bayesian inference algorithm are sensitive to the selected hyper-parameters [14, 12]. For example Fig.2(a) illustrates a poor quality segmentation that results from poorly selected hyper-parameters.

### 3.1 Hypothesis Tests for Model Consistency with Segments

To ensure model consistency we must test if each segment in $\bar{\mathcal{D}}$ is consistent with a multivariate Gaussian process (i.e. samples are independent and normally distributed). To test if the segment $\{\boldsymbol{y}_t\}_{t \in T_k} \in \bar{\mathcal{D}}$ contains independent samples we evaluate the autocorrelation function (ACF) [5] for each segment. For $\{\boldsymbol{y}_t\}_{t \in T_k}$ the ACF must exponentially decay to zero which indicates that the segment contains independent samples. Note that it is possible for a spurious autocorrelation structure to be present in the segment if the segment is composed of a mixture of Gaussian processes. If this is suspected then the hyper-parameters of the non-parametric Bayesian inference algorithm are updated to increase the number of segments (for example by increasing $L$ or decreasing $\kappa$). Since there is no universally most powerful test for multivariate normality, we use the improved Bonferroni method [23] which contains four affine invariant hypothesis test statistics elevating the need to select the most sensitive single test while retaining the benefits of the these four multivariate normality tests.

### 3.2 Data-Driven Confidence Bounds for Dynamic Model Estimation

An important consideration when evaluating the quality of the segmentation $\bar{\mathcal{D}}$ is that each segment contains sufficient samples to confidently estimate the mean and covariance $\{\boldsymbol{\mu}, \Sigma\}$ of the SMG model. This is particularly important in medical applications as it provides an estimate of the maximum number of samples needed to confidently estimate $\{\boldsymbol{\mu}, \Sigma\}$ which are used to estimate the clinical state of the patient. Note that the estimated posterior distribution for $\{\boldsymbol{\mu}, \Sigma\}$ can not be used to bound the number of samples required. To estimate $\{\boldsymbol{\mu}, \Sigma\}$ given $\{\boldsymbol{y}_t\}_{t \in T_k}$, the maximum likelihood estimators given by:

$$\hat{\boldsymbol{\mu}}(k) = \frac{1}{n_k} \sum_{t=1}^{n_k} \boldsymbol{y_t}, \quad \hat{\Sigma}(k) = \frac{1}{n_k} \sum_{t=1}^{n_k} (\boldsymbol{y}_t - \hat{\boldsymbol{\mu}}(k))(\boldsymbol{y}_t - \hat{\boldsymbol{\mu}}(k))' \tag{7}$$

are used with $n_k = |T_k|$ is the total number of samples in segment $k \in \mathcal{K}$. If each vital sign is independent (i.e. spherical multivariate Gaussian distribution) then an empirical Bernstein bound [13] can be constructed to estimate the error between the sample mean $\hat{\boldsymbol{\mu}}$ and the actual mean $\boldsymbol{\mu}$. From the empirical Bernstein bound, the minimum number of samples necessary to ensure that $P(\hat{\mu}(k, j) - \mu(k, j) \geq \varepsilon) \leq \alpha$ for all segments $k \in \mathcal{K}$ and streams $j \in \{1, \ldots, m\}$ for some confidence level $\alpha > 0$ and tolerance $\varepsilon \geq 0$ is given by:

$$n(\varepsilon, \alpha) \geq \left( \frac{6\sigma_{\max}^2 + 2\Delta_{\max}\varepsilon}{3\varepsilon^2} \right) \ln\left(\frac{1}{\alpha}\right) \tag{8}$$

with $\sigma_{\max}^2$ the maximum possible variance and $\Delta_{\max}$ the maximum possible difference between the maximum and minimum values of all values in the vital sign data.

To construct a relaxed bound on the sample mean $\hat{\boldsymbol{\mu}} \in \mathbb{R}^m$, and a bound on the sample covariance $\hat{\Sigma} \in \mathbb{R}^{m \times m}$ computed using (7), we generalize the empirical Bernstein bound to the multidimensional case. The goal is to construct a bound of the form $P(||Z|| \geq \varepsilon) \leq \alpha$ where $|| \cdot ||$ denotes the spectral norm if $Z$ is a matrix, or the 2-norm in the case $Z$ is a vector. To construct a probabilistic bound on the accuracy of the estimated mean we utilize the vector Bernstein inequality given by Theorem 1.

**Theorem 1** *Let $\{Y_1, \ldots, Y_n\}$ be a set of independent random vectors with $Y_t \in \mathbb{R}^m$ for $t \in \{1, \ldots, n\}$. Assume that each vector has uniform bounded deviation such that $||Y_t|| \leq L \quad \forall t \in \{1, \ldots, n\}$. Writing $Z = \sum_{t=1}^{n} Y_t$, then*

$$P(||Z|| \geq \varepsilon) \leq (2m) \exp\left(\frac{-3\varepsilon^2}{6V(Z) + 2L\varepsilon}\right), \quad V(Z) = \sum_{t=1}^{n} E[||Y_t||_2^2]. \tag{9}$$

The proof of Theorem 1 is provided in the Supporting Material. To construct the bound on the number of samples necessary to estimate the mean we define $Z = \hat{\boldsymbol{\mu}} - \boldsymbol{\mu}$ with $Y_t = (\boldsymbol{y}_t - \boldsymbol{\mu})/n$. Using the triangle inequality, Jensen's inequality, and assuming $||\boldsymbol{y}_t||_2 \leq B_1$ for some constant $B_1$, we have that:

$$L \leq \frac{2B_1}{n}, \qquad V(Z) \leq \frac{1}{n}\left(B_1^2 - ||\boldsymbol{\mu}||_2^2\right). \tag{10}$$

Plugging (10) into (9) results in the minimum number of samples necessary to guarantee that $P(||\hat{\boldsymbol{\mu}} - \boldsymbol{\mu}|| \geq \varepsilon) \leq \alpha$ with the number of samples $n(\varepsilon, \alpha)$ given by:

$$n(\varepsilon, \alpha) \geq \left(\frac{6(B_1^2 - ||\boldsymbol{\mu}||_2^2) + 4B_1\varepsilon}{3\varepsilon^2}\right) \ln\left(\frac{2m}{\alpha}\right). \tag{11}$$

To bound the number of samples necessary to estimate $\Sigma$ we utilize the corollary of Theorem 1 for real-symmetric matrices with $Z = \hat{\Sigma} - \Sigma$. The bound on the number of samples necessary to guarantee $P(||\hat{\Sigma} - \Sigma|| \geq \varepsilon) \leq \alpha$, assuming $||\Sigma|| \leq ||\boldsymbol{y}_t - \hat{\boldsymbol{\mu}}|| \leq B_2$, is given by:

$$n(\varepsilon, \alpha) \geq \left(\frac{6B_2^2 + 4B_2\varepsilon}{3\varepsilon^2}\right) \ln\left(\frac{2m}{\alpha}\right). \tag{12}$$

For a given $\alpha$ and $\varepsilon$, and an estimate of the maximum spectral norm of $\Sigma$ and norm of $\boldsymbol{\mu}$, equations (11) and (12) can be used to estimate the minimum number of samples necessary to sufficiently estimate $\{\boldsymbol{\mu}, \Sigma\}$. To accurately compute the clinical state from the unique dynamic model, each segment must satisfy (11) and (12), otherwise any clinical state estimation may give unreliable results.

### 3.3 Statistical Tests for Statistically Identical Dynamic Models

In this section we construct a novel hypothesis test for mean and covariance equality with a given confidence, and design parameters that control the importance of the mean equality compared to the covariance equality. The hypothesis test both evaluates the quality of the estimated stochastic model, but can also be used to merge statistically identical segments to increase the accuracy of the dynamic model parameter estimates. Given two segments of vital signs, each associated with a supposedly unique dynamic model, we define the null hypothesis $H_0$ as the equality of the mean and covariance matrices from the two dynamic models, and the alternate hypothesis $H_1$ that either the mean or covariance are not equal. Formally:

$$H_0 : \Sigma(k) = \Sigma(k') \text{ and } \boldsymbol{\mu}(k) = \boldsymbol{\mu}(k'), \quad H_1 : \Sigma(k) \neq \Sigma(k') \text{ or } \boldsymbol{\mu}(k) \neq \boldsymbol{\mu}(k'). \tag{13}$$

Several methods exist for testing for covariance equality [20] and for mean equality [24], however we wish to test for both covariance and location equality. To test for the global hypothesis $H_0$ in (13), note that $H_0$ and $H_1$ can equivalently be stated as a combination of the sub-hypothesis as follows:

$$H_0 : H_0^1 \cap H_0^2 \quad \text{and} \quad H_1 : H_1^1 \cup H_1^2 \tag{14}$$

with $H_0^1 : \boldsymbol{\mu}(k) = \boldsymbol{\mu}(k')$, $H_1^1 : \boldsymbol{\mu}(k) \neq \boldsymbol{\mu}(k')$, $H_0^2 : \Sigma(k) = \Sigma(k')$, and $H_1^2 : \Sigma(k) \neq \Sigma(k')$. To construct the hypothesis test for $H_0$ the non-parametric the permutation testing method [17] is used which allows us to combine the sub-hypothesis tests for covariance and mean equality to construct a hypothesis test for $H_0$.

To test for the null hypothesis $H_0^1$ we utilize Hotelling's $T^2$ test as it is asymptotically the most powerful invariant test when the data associated with $k$ and $k'$ are normally distributed [4]. Given that

$\boldsymbol{y}_t$ are generated from a multivariate normal distribution, the test statistic $\tau^1$ follows a $T^2$ distribution such that $\tau^1 \sim T^2(m, n(k) + n(k') - 2)$ where $n(k)$ and $n(k')$ are the number of samples in segments $k$ and $k'$ respectively. To test for the null hypothesis $H_0^2$ we utilize the modified likelihood ratio statistic provided by Bartlett [1], written $\Lambda^*$, which is uniformly the most power unbiased test for covariance equality [15]. The test statistic for covariance equality is given by:

$$\tau^2 = -2\rho \log(\Lambda^*), \quad \rho = 1 - \frac{2m^2 + 3m - 1}{6(m+1)n}(n/n(k) + n/n(k') - 1), \quad n = n(k) + n(k').$$

From (Theorem 8.2.7 in [15]) the asymptotic cumulative distribution function of $\tau^2$ can be approximated by a linear combination of $\chi^2$ distributions which has a convergence rate of $O((\rho n)^{-3})$.

To construct the permutation test for $H_0$ Tippett's combining function [17] is used with $H_0$: $\tau = \min(\lambda^1/k^1, \lambda^2/k^2)$ where $\lambda^1$ and $\lambda^2$ are the p-values of the sub-hypothesis tests $H_0^1$ and $H_0^2$ respectively, and $k^1$ and $k^2$ are design parameters. If $k^1 > k^2$ then the mean equality is weighted more then the covariance equality. If $k^1 = k^2$ then both mean equality and covariance equality are weighted equally. For the test statistics $\tau^1$ and $\tau^2$ the p-values are given by $\lambda^1 = P(\tau^1 \geq \tau_0^1)$ and $\lambda^2 = P(\tau^2 \geq \tau_0^2)$ where $\tau_0^1$ and $\tau_0^2$ are realizations of the test statistics. To utilize $\tau$ as a test statistic we require the cumulative distribution function of $\tau$. Note that if $H_0^1$ is true (i.e. mean equality) then the distributions of $\tau^1$ and $\tau^2$ are independent since $\tau^1$ follows a $T^2$ distribution which results in $\lambda^1 \sim \mathcal{U}(0,1)$ and $\lambda^2 \sim \mathcal{U}(0,1)$ [17]. The cumulative distribution function of $\tau$ is given by $P(\tau \leq x) = (k^1 + k^2)x - k^1 k^2 x^2$ for $x \in [0, \min(1/k^1, 1/k^2)]$. Given $P(\tau \leq x)$, for a significance level $\alpha$, we reject the null hypothesis $H_0$ if $\tau \leq \delta$ where $\delta$ is the solution to $P(\tau \leq \delta) = \alpha$. The parameter $\delta$ is given by: $\delta = \left((k^1 + k^2) - \sqrt{(k^1 + k^2)^2 - 4\alpha k^1 k^2}\right)/(2k^1 k^2)$.

For a given significance level $\alpha$, and design parameters $k^1$ and $k^2$, we can test $H_0$ for the samples $\{\boldsymbol{y}_t\}_{t \in T_k}$ and $\{\boldsymbol{y}_t\}_{t \in T_{k'}}$ by evaluating $\tau_0 = \min(\lambda_0^1/k^1, \lambda_0^2/k^2)$ with $\lambda_0^1$ and $\lambda_0^2$ the realizations of the p-values for $\tau_1$ and $\tau_2$. By repeatedly applying this hypothesis test to segments $\{\boldsymbol{y}_t\}_{t \in T_k}$ for $k \in \mathcal{K}$ we can detect any segments with equal mean and covariance with a significance level $\alpha$. Similar segments can be merged to increase the accuracy of the estimated dynamic model parameters, or be used to evaluate the quality of the patient's stochastic model.

## 4    Estimating Patient's Clinical State using Clinician Domain-Knowledge

In this section the Algorithm 1 (Fig.1) is presented which constructs stochastic models of patients based on their historical EHR data and clinician domain-knowledge, and is used to classify the clinical state of new patients.

Algorithm 1 is composed of five main steps. Step#1 to Step#2 are used to construct the stochastic models of the patients based on the EHR data $\mathcal{D}$, and to construct the segmented dataset $\bar{\mathcal{D}}$ (1). The stochastic models are constructed using the non-parametric Bayesian inference algorithm from Sec.2. Step#2 measures the quality of the stochastic models, and iteratively updates the hyper-parameters of the Bayesian inference algorithm to guarantee the quality of the detected dynamic models as discussed in Sec.3. In Step#3 each segment (e.g. dynamic model) in $\bar{\mathcal{D}}$ is labelled by the clinician, based on the clinical states of interest, to construct the dataset $\mathcal{L}$. Step#4 and Step#5 involves the online portion of the algorithm which constructs stochastic models for new patients and estimates their clinical state based on each patient's estimated stochastic model. Step#4 constructs the stochastic model for the new patient, then in Step#5 each unique dynamic model from Step#4 is associated with a clinical state of interest using the labelled dataset $\mathcal{L}$ from Step#3. Note that $\mathcal{L}$ contains several segments (e.g. dynamic models) that are associated with one clinical state. To estimate the clinical state of the new patient a similarity metric based on the Bhattacharyya distance, written $D_B(\cdot)$, is used. If the minimum Bhattacharyya distance between the new patients segment $k$ and next closest segment $k' \in \mathcal{L}$ is greater then $\delta_{\text{th}}$ the segment is labelled as anomalous, otherwise the segment is given the label of segment $k' \in \mathcal{L}$. Information on the computational complexity and implementation details of Algorithm 1 are provided in the Supporting Material.

## 5    Real-World Clinical State Estimation in Cancer Ward

In this section Algorithm 1 is applied to a real-world EHR dataset composed of a cohort of patients admitted to a cancer ward. A detailed description of the dataset is provided in the Supporting Material.

**Algorithm 1** Patient Clinical State Estimation

Step#1: Construct stochastic models for each patient using $\mathcal{D}$ and the non-parametric Bayesian algorithm presented in Sec.2. Using the stochastic models construct the dataset $\bar{\mathcal{D}}$ (1).

Step#2: To evaluate the quality of each stochastic model, each segment in $\bar{\mathcal{D}}$ from Step#1 is tested for: i) model consistency, ii) sufficient samples to guarantee accuracy of dynamic model parameter estimates, and iii) statistical uniqueness of segments using the methods in Sec.3. If the quality is not sufficient then return to Step#1 with updated hyper-parameters for the non-parametric Bayesian inference algorithm.

Step#3: Given $\bar{\mathcal{D}}$ and the clinical states of interest, the clinician constructs the labelled dataset $\mathcal{L} = \{(\{\boldsymbol{y}_t^i\}_{t \in T_k^i}, l_k^i), k \in \{1, \dots, K^i\} = \mathcal{K}^i\}$.

Step#4: For a new patient $i = 0$ with vital signs $\{\boldsymbol{y}_t^0\}_{t \in T^0}$, construct the stochastic model of the patient using the Bayesian non-parametric learning algorithm. Then, based on the stochastic model, construct the segmented vital sign data $\{\{\boldsymbol{y}_t^0\}_{t \in T_k^0}, k \in \{1, \dots, K^0\} = \mathcal{K}^0\}$.

Step#5: To estimate the label $l(k)$, written $\hat{l}(k)$, of each segment $k \in \mathcal{K}^0$ from Step#4, compute the solution to the following optimization problem for each $k$:

$$\text{if } \min_{l \in \mathcal{L}}\{D_B(k, k')\} \geq \delta_{\text{th}} \text{ then } \hat{l}(k) = \emptyset, \text{ else } \hat{l}(k) \in \underset{l \in \mathcal{L}}{\text{argmin}} \left\{ \frac{\min_{k' \in \mathcal{L}_l}\{D_B(k, k')\}}{\min_{k' \in \mathcal{L}_{-l}}\{D_B(k, k')\}} \right\}$$

with $\emptyset$ the anomalous state, $\mathcal{L}_l \in \mathcal{L}$ the set of segments that are labeled with $l$, $\mathcal{L}_{-l} \in \mathcal{L}$ the set of all segments that are not labeled as $l$, and $\delta_{\text{th}}$ is a threshold. Return to Step#4.

The first step of Algorithm 1 is to segment the EHR data based on the estimated stochastic models of the patients. Fig.2(a) illustrates the dynamic models of a specific patient's estimated stochastic model for $\kappa = 0.1$ and $S_0 = 0.1 I_m$ ($I_m$ is the identity matrix), and for $\kappa = 1$ and $S_0 = I_m$. As seen, for $\kappa = 0.1$ and $S_0 = 0.1 I_m$ several segments have insufficient samples for estimating the model parameters, and are not statistically unique. However the segments resulting from $\kappa = 1$ and $S_0 = I_m$ provide a stochastic model of sufficient quality where each segment contains sufficient samples to accurately estimate the model parameters, the segments are statistically unique, and satisfy the multivariate normality assumption. Therefore we set $\kappa = 1$ and $S_0 = I_m$ to construct the segmented dataset $\bar{\mathcal{D}}$ from $\mathcal{D}$. The dataset $\mathcal{L}$ is constructed by providing the clinician with $\bar{\mathcal{D}}$ who then labels each segment as either in the ICU admission clinical state, or non-ICU clinical state.

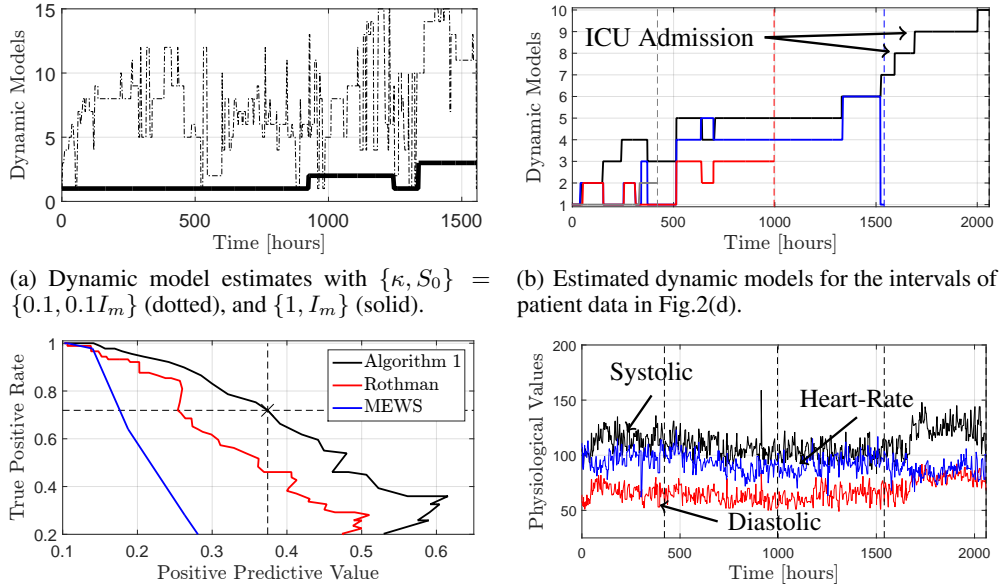

(a) Dynamic model estimates with $\{\kappa, S_0\} = \{0.1, 0.1 I_m\}$ (dotted), and $\{1, I_m\}$ (solid).

(b) Estimated dynamic models for the intervals of patient data in Fig.2(d).

(c) Trade off between the TPR and PPV. The dashed cross-hair indicates the performance of Algorithm 1 for $\delta_b = 1$.

(d) Physiological signals from the patient with discovered models in Fig.2(b).

Figure 2: Dynamic model discovery and performance of Algorithm 1.

Of critical importance in medical applications is the accuracy and timeliness of the detection of the clinical state of the patient. Fig.2(b) provides the trade-off between the TPR and PPV between Algorithm 1, Rothman index [18] which is a state-of-the-art method utilized in many hospitals today, and MEWS [21] which are dependent on the threshold selected for each. As seen Algorithm 1 has a superior performance compared to these two popular risk scoring methods. For example if we require the TPR = 71.9%, then the associated PPV values for the Rothman index and MEWS are 26.1% and 18.0% respectively. There is a 11.3% increase in the PPV value for the Rothman index, and 19.4% increase in the PPV for MEWS compared to the PPV of Algorithm 1. We also compare with methods commonly used in medical with the results presented in Table 1. As seen, Algorithm 1 outperforms all these methods for estimating the patient's clinical state. There are several possible reasons that Algorithm 1 outperforms these methods including accounting for therapeutic interventions and utilizing fine-grained personalization. Note that the results in Table 1 are computed 12 hours prior to ICU admission or hospital discharge. Additionally, the average detection time of ICU admission or discharge using Algorithm 1 is approximately 24 hours prior to the clinician's decision. This timeliness ensures that the patient's clinical state estimate provides clinicians with sufficient warning to apply a therapeutic intervention to stabilize the patient.

Table 1: Accuracy of Methods for Predicting ICU Admission

| Algorithm | TPR(%) | PPV(%) |
|---|---|---|
| Algorithm 1 | 71.9% | 37.4% |
| Rothman Index | 53.9% | 34.5% |
| MEWS | 28.1% | 26.3% |
| Logistic Regression | 55.7% | 30.7% |
| Lasso Regularization | 55.8% | 30.3% |
| Random Forest | 44.5% | 31.1% |
| SVMs | 32.2% | 29.9% |

A key feature of Algorithm 1 is that it learns the number of unique dynamic models for each patient, and as more data is collected the number of unique dynamic models discovered may increase. Fig.2(b) illustrates this process for a patient with associated physiological signals given in Fig.2(d). The horizontal dashed line indicates the intervals and associated discovered dynamic models. Note that typical hospitalization time for cancer ward patients in the dataset range from 4 hours to over 85 days. As seen, as more samples are obtained for the patient the number of dynamic models that describe the patient's dynamics increase. Additionally, there is good agreement between where the patient's dynamics change for the different time intervals. For example the change point at 40 hours after hospitalization occurs as a result of an increase in the systolic and diastolic blood pressure, and a decrease in the heart-rate. At 1700 hours the change in state results from a dramatic increase in both the systolic and diastolic blood pressure, and a decrease in the heart-rate. From Fig.2(d) these physiological signals were not observed previously, therefore Algorithm 1 correctly detects that this is a new unique state for the patient. Though Algorithm 1 can identify changes in patient state, the domain-knowledge from the clinician is required to define the clinical state of the patient. Only dynamic models 8 and 9 are associated with the ICU admission state.

Further results are provided in the Supporting Material that illustrate how current methods for constructing risk scores suffer from the bias introduced from therapeutic intervention censoring, and how a binary threshold $\delta_b$ can be introduced into Algorithm 1 for controlling the TPR and PPV for clinical state estimation.

## 6 Conclusion

In this paper a novel non-parametric learning algorithm for confidently learning stochastic models of patient's and classifying their associated clinical state was presented. Compared to state-of-the-art clinical state estimation methods our algorithm eliminates the bias caused by therapeutic intervention censoring, is personalized to the patient's specific dynamics resulting from medical complication (e.g. disease, drug interactions, physical contusions or fractures), and can detect anomalous clinical states. The algorithm was applied to real-world patient data from a cancer ward in a large academic hospital, and found to have a significant improvement in classifying patient's clinical state in both accuracy and timeliness compared with current state-of-the-art methods such as the Rothman index. The algorithm provides valuable information to allow clinicians to make informed decisions about selecting if a therapeutic intervention is necessary to improve the clinical state of the patients.

**Acknowledgments**

This research was supported by: NSF ECCS 1462245, and the Airforce DDDAS program.

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
