[Supplementary Material · nips_2016_sm.pdf]

# Supporting Material:
# A Non-parametric Learning Method for Confidently Estimating Patient's Clinical State and Dynamics

**William Hoiles**
Department of Electrical Engineering
University of California Los Angeles
Los Angeles, CA 90024
whoiles@ucla.edu

**Mihaela van der Schaar**
Department of Electrical Engineering
University of California Los Angeles
Los Angeles, CA 90024
mihaela@ee.ucla.edu

## S1   Related Work

**Risk Scores**

There are two main methods to compute the risk scores of patients: expert-domain knowledge and classification. Popular expert-domain based risk score measures include the Acute Physiological and Chronic Health Evaluation (APACHE) [17], and modified Early Warning Score (MEWS) [25]. Both the APACHE and MEWS score are designed by having the experts define the risk based on the physiological stream values. A limitation with these methods is that they have a poor positive predictive value and a high number of false positives in the range of 70-95% [1]. The classification approach learns a mapping between electronic health record data and the morbidity and mortality of the patients. For example the Rothman index [22] provides a continuous measure of the patients risk which is computed using the mortality rate of patients after 1-year and is dependent on the vital signs, lab results, cardiac rhythms and nursing assessments of the patient. A limitation with these methods is that they contain a bias as a result of therapeutic intervention censoring as they do not account for therapeutic interventions. Additionally, these methods are not personalized as they are trained without considering patient's with different medical conditions, demographics, and the unique therapeutic interventions applied to each patient. As such, a one size fits all classifier is expected to have a lower accuracy compared to our algorithm (Fig.1) which accounts for therapeutic interventions and is personalized. Using real-world data from a cancer ward in a large academic hospital we illustrate how our algorithm significantly improves the accuracy of estimating the clinical state of the patient compared to these popular risk scoring methods.

**Probabilistic Generative Models of Patient's Physiological Signals and State**

Dynamic models that can succinctly capture the generative model of the measured physiological data as a function of the underlying patient state provides vital information for both classifying the patient state and for removing the bias introduced from therapeutic intervention censoring. A common method is to utilize multistate models based on Markov processes (e.g. hidden Markov models) for modeling the state changes of the patient. Once the HMM parameters are estimated, using methods such as the Baum-Welch algorithm, then the hidden state of the HMM can be computed by inference of the HMM model. This technique has been applied to estimate several time-varying patient states for liver cancer [16], breast cancer [19], bronchiolitis obliterans syndrome [14], HIV [8], Alzheimer's disease [6], hepititis C [26], and abdominal aortic aneurysms [15]. In the HMM each discrete state variable determines the momentary state of the patient. However, there are two important limitations with applying the HMM to estimate patient state. First the typical maximum likelihood estimation procedures to estimate the model parameters may introduce over-fitting and under-fitting as mode

complexity (i.e. the necessary number of states) is not accounted for. Second, the model structure has to be defined a priori.

To mitigate the limitations of the HMM requires a richer class of stochastic processes known as combinatorial stochastic processes in which we assume an infinite number of possible states. Intuitively this generative model will have unbounded complexity such that under fitting is mitigated, and the effects of over-fitting can be reduced using the tools of non-parametric Bayesian inference [29]. Additionally, utilizing combinatorial stochastic processes as the patient's generative model allows learning new states as further patient data is collected. Note that non-parametric here does not mean a parameter-less model, but rather a model in which the number of states grows as more data is observed for the patient. Common combinatorial stochastic models for time-series segmentation include the Hierarchical Dirichlet process (HDP) [30], Beta Process [12, 31], and Pitman-Yor process [13]. The HDP can be interpreted as a non-parametric generalization of the HMM to an infinite state HMM where the number of states is unbounded and can be learned from the data. The HDP assumes that all time series share the same set of behaviors and switch among them in exactly the same manner. Application examples of where the HDP is utilized to segment temporal data can be found in [7, 30, 2]. An example application for the HDP for segmenting clinical temporal data is provided by Saria et al. [23] which utilizes the HDP to cluster data in electronic health records for natural language processing. A major limitation with blindly applying non-parametric Bayesian inference to estimate the generative model parameters of the patient is that for small sample sizes $n$ the parameter estimates are sensitive to the selected prior distributions. If these priors place little weight on the true parameter values this will result in a poor segmentation. Additionally, the rate of convergence of the posterior distribution for infinite-dimensional models to the true posterior distribution is typically $O(1/\sqrt{n})$ or slower [9, 11, 10, 18]. To mitigate these issues our algorithm ensures that the resulting segmentation is consistent with the modeling assumptions, that each segment contains sufficient samples for parameter estimation of the dynamic models, and that each segment is statistically unique from the other detected segments. If the segmentation from the non-parametric Bayesian inference algorithm fails to produce a valid segmentation, then the prior parameters or/and modeling assumptions must be adjusted.

**Novelty Detection in Physiological Data**

Novelty detection or anomaly detection is typically defined as the detection of a unique set of physiological signals that are not contained in the dataset $\mathcal{D}$ used for training. There are four main classes of anomaly detection techniques that are used to detect for patients with unique physiological signals: probabilistic, distance-based, reconstruction based, and domain based [20]. Algorithm 1 utilizes a distance-based measure for detecting anomalous physiological signals. In [5] a support-vector machine is trained for the one-class problem where a patient is either in the "normal" or "abnormal" state. In [21] a factorial switching Kalman filter model with a priori states associated with normal and abnormal states is defined. A likelihood function is then utilized to estimate if the patient has entered an abnormal state. In [24] each patient's physiological data is fitted to a Gaussian distribution, then Horn's algorithm is utilized to detect any data that is not consistent with the estimated distribution. In [28] a k-means clustering algorithm is used to construct prototype physiological patterns, then for each pattern computes the Parzen-Window density estimation. Any patient data not consistent with the estimated distributions are classified as abnormal. The main limitation of these methods is that they are not personalized for patient's and diseases, they are biased as a result of therapeutic intervention censoring, and the number of novel states must be defined a priori. Our algorithm for detecting novelty overcomes all these limitations as it utilizes fine-grained personalization, does not contain bias resulting from therapeutic interventions, and learns the number of segments non-parametrically.

## S2   Cancer Ward Dataset Description

The cohort comprises 1065 patients who were diagnosed with leukemia, lymphoma, multiple myeloma and other hematologic malignancies. The patients were receiving chemotherapy, allogeneic stem cell transplantation, or autologousstem cell transplantation during their hospitalization in the cancer ward. All patients are in-patients (i.e. they where in the hospital for the duration of the analysis unless discharged). The therapeutic treatments received by patients may cause severe immunosuppression during their hospitalization placing them at an extreme risk of clinical deterioration,

Table 1: Properties of the Cancer Ward Patient's Vitals

| Parameter | Mean | Standard Deviation | Minimum | Maximum |
|---|---|---|---|---|
| Systolic Blood Pressure (mmHg) | 122.45 | 16.83 | 54.00 | 243.00 |
| Diastolic Blood Pressure (mmHg) | 73.48 | 11.66 | 3.00 | 144.00 |
| Heart Rate (beats per minute) | 84.49 | 16.99 | 0.00 | 237.00 |
| Respiratory Rate (breaths per minute) | 18.82 | 2.67 | 0.00 | 180.00 |
| Temperature (F) | 98.45 | 1.13 | 32.00 | 106.90 |
| Peripheral Capillary Oxygen Saturation (%) | 96.76 | 8.45 | 0.00 | 100.00 |
| Haemoglobin (g/dL) | 5.49 | 18.16 | 0.00 | 364.19 |
| White Blood Cell Count ($\times 10^{-3}$/mL) | 9.26 | 1.39 | 3.60 | 17.20 |
| Platelet Count ($\times 10^{-3}$/mL) | 70.63 | 85.77 | 1.00 | 772.00 |
| Sodium Concentration (mmol/L) | 137.32 | 3.51 | 107.00 | 154.00 |
| Potassium Concentration (mmol/L) | 3.89 | 0.50 | 2.00 | 9.30 |
| Chloride Concentration (mmol/L) | 104.19 | 4.35 | 73.00 | 124.00 |
| Total Carbon Dioxide (mEq/L) | 24.89 | 3.20 | 10.00 | 45.00 |
| Blood Urea Nitrogen (g/dL) | 16.72 | 12.99 | 2.00 | 153.00 |
| Creatinine (mg/dL) | 0.97 | 1.12 | 0.10 | 19.10 |
| Glucose (mg/dL) | 120.37 | 42.86 | 39.00 | 801.00 |

which requires ICU admission. The number of patients admitted to ICU is 101, which comprises 9.48% of the 1065 patents in the cohort. Each patient's electronic health record is associated with 17 temporal physiological data streams with the vital signs (systolic blood pressure, diastolic blood pressure, heart-rate, respiratory-rate, temperature, etc.), and laboratory tests (white blood cell count, haemoglobin, glucose, sodium concentration, potassium concentration, chloride concentration, etc.). Table 1 provides the vital signs and laboratory tests used for analysis and their associated properties. A representative example of the physiological values and associated discovered segments are provided in Fig.S1. Note that as a result of patient anonymity we do not have demographic information (e.g. age, sex, comorbidity) for the patients. The sampling rate of the vital signs is approximately every 4 hours, and the sampling rate of the laboratory tests are approximately every 24 hours. Note that as these sampling rates the physiological data is not expected to have significant autocorrelations present. The interval of each patient's hospitalization until either discharge or ICU admission varies across patients and is not known a priori. Given the unique medical conditions of each patient, we can not define precise values of the physiological signals that are associated with the ICU admission state for all patients. Instead, Algorithm 1 learns the fine-grained personalized values applicable for the patients to detect if they have entered the ICU admission clinical state.

Figure S1: Physiological signals from the patient with discovered dynamic models in Fig.2(b). Note the coloured segments are for illustration only, these colors are not related to the specific physiological values.

# S3    Algorithm 1: Therapeutic Intervention Censoring and Risk Scores

In this section we illustrate the similarity and differences of Algorithm 1 compared to the Rothman index [22] and MEWS [25]. Specifically we compare the performance of these methods based on the resulting confusion matrices for different threshold values, and illustrate how Algorithm 1 mitigates therapeutic intervention censoring compared to these popular risk scoring methods.

To gain insight into the similarities and differences of Algorithm 1, Rothman index, and MEWS, Fig.S2 compares the state estimate from Algorithm 1, Rothman index, and MEWS for four patients. For the Rothman and MEWS risk scores we must set the threshold for the ICU admission state. For the Rothman index we set the threshold at 0.65, and for MEWS we set the threshold at 0.5. The associated accuracy for these thresholds is provided in Table 1. As seen in Fig.S2(a) and Fig.S2(b) the results of Algorithm 1 are in agreement with the results of the Rothman index and MEWS risk score. Notice that in Fig.S2(b) both Algorithm 1 and the Rothman index detect that the patient is in the ICU admission state at 625 hours, MEWS at 850 hours, and the Rothman index at 900 hours. However, as a result of therapeutic intervention neither the Rothman index or MEWS scores detect the ICU admission state at 175 hours. In Fig.S2(c) we see that the results of Algorithm 1 have a faster timeliness for detecting the ICU admission state compared with the Rothman and MEWS. The timeliness of detecting ICU admission is vital to ensure that clinicians have sufficient time to perform therapeutic interventions to attempt to transition the patient out of the ICU admission state. In Fig.S2(d) we see that Algorithm 1 is able to sufficiently estimate that patient's clinical state of ICU admission while both the Rothman index and MEWS do not detect the patient has entered the ICU admission state–again illustrating the effects therapeutic intervention censoring, as in the case of risk scores, this would not be considered an ICU admission state as the patient recovered as a result of a therapeutic intervention. This can result significant patient harm and cost as these risk scores recommend the patient's physiological condition is improving when in fact it is worsening.

S2(a) Agreement with the Rothman index and MEWS index for the detection of the ICU admission state.

S2(b) Partial agreement (for dynamic model 5) with the Rothman index and MEWS index for the detection of the ICU admission state.

S2(c) Agreement with the Rothman index and MEWS index for the detection of the ICU admission state.

S2(d) Disagreement with the Rothman index and MEWS index for the detection of the ICU admission state.

Figure S2: The estimated clinical states from Algorithm 1 (black), the Rothman index (red), and MEWS (blue) computed from typical physiological signals from the patients. The dotted horizontal black line indicates the threshold for the Rothman index, and the dotted gray line indicates the threshold for MEWS to indicate that the patient has entered the ICU admission state.

The performance of Algorithm 1, the Rothman index, and MEWS are dependent on the selected threshold value for ICU admission which should balance the performance metrics true positive rate (TPR), positive predictive value (PPV), false positive rate (FPR), and false negative rate (FNR).

Fig.S3 illustrates how these performance metrics change for different threshold values. The TPR, PPV, FPR, and FNR of Algorithm 1 are respectively: 71.9%, 37.4%, 28.1%, and 13.7% for $\delta_b = 1$. From Fig.reffig:rothmanANDmewsacc we see that Algorithm 1 has a superior performance compared to the Rothman index and MEWS. For example if we require the TPR = 69.9%, then the associated PPV values for the Rothman index and MEWS are 26.1% and 18.0% respectively. There is an 11.3% increase in the PPV value for the Rothman index, and 19.4% increase in the PPV for MEWS compared to the PPV of Algorithm 1. In clinical setting it is vital to keep the FNR and FPR low to ensure patient's in the ICU admission state are correctly identified while ensuring the number of false alarms is low. For an FNR = 13.7% the Rothman index has an FPR of 43.3% which is an increase in false alarms of 15.2% compared with Algorithm 1, and the MEWS has an FPR of 64.4% which has an increase in false alarms of 36.3% compared to Algorithm 1. Therefore Algorithm 1 has a significantly reduced false alarm rate compared with the Rothman index and MEWS while maintaining a sufficiently low FNR.

S3(a) Trade off between the true positive rate (TPR) and positive predictive value (PPV) for Algorithm 1, Rothman index, and MEWS. The dashed crosshair indicates the performance of Algorithm 1 for $\delta_b = 1$.

S3(b) Trade off between the false positive rate (FPR) and false negative rate (FNR) for Algorithm 1, Rothman index, and MEWS. The dashed cross-hair indicates the performance of Algorithm 1 for $\delta_b = 1$.

Figure S3: The estimated performance of Algorithm 1 (black), the Rothman index (red), and MEWS (blue) for different threshold values.

## S4    Selection of the Threshold $\delta_b$ for Algorithm 1

In this section we study how the threshold $\delta_b$ can be introduced into Algorithm 1 to ensure a sufficient balance between the performance metrics TPR and PPV. Note that $\delta_b$ is similar to the thresholding values that must be selected for the Rothman index [22] and MEWS [25] when classifying if the patient has entered the ICU admission clinical state. The two clinical states we consider in this paper are discharge or ICU admission which we can write as $l \in \{\text{DIS}, \text{ICU}\}$. To compute a measure of the segment $k \in \mathcal{K}$ being in the DIS clinical state we utilize:

$$\delta_{\text{DIS}}(k) = \frac{\min_{k' \in \mathcal{L}_{\text{DIS}}}\{D_B(k, k')\}}{\min_{k' \in \mathcal{L}_{\text{ICU}}}\{D_B(k, k')\}}. \tag{S1}$$

which is the entry in the optimization problem in Step#5 of Algorithm 1. It is clear that if $\delta_{\text{DIS}}(k) > 1$ then the clinical state is associated with ICU admission, and if $\delta_{\text{DIS}}(k) < 1$ then the clinical state is associated with the DIS clinical state. Therefore we set the threshold value of Algorithm 1 to $\delta_b = 1$ such that if $\delta_{\text{DIS}}(k) > \delta_b$ then the patient is in the ICU admission state. Fig.S4 illustrates the values of $\delta_{\text{DIS}}(k)$ for all dynamic models detected with Fig.S4(a) including all the dynamic models that do not include a clinician defined state, and Fig.S4(b) with dynamic models that have a clinician defined state. Formally, for any segmentation $\{\{\boldsymbol{y}\}_{t \in T_k^i}, k \in \mathcal{K}^i\}$ for patient $i \in \mathcal{I}$ with associated clinician defined clinical state $l_{t'}^i$ at time $t'$, only segments that satisfy $t' \notin T_k^i$ are in Fig.S4(a), and only segments that satisfy $t' \in T_k^i$ are in Fig.S4(b). As seen from the results in Fig.S4(a), several of the patients that are eventually discharged from the hospital enter the ICU admission state during their hospitalization period. In risk scoring methods these would be considered as false positives, however this type of assumption is what leads to the bias resulting from therapeutic intervention censoring. The only time a false positive (patient is in discharge state however Algorithm 1 recommends ICU admission) is detected is if the dynamic model that includes the clinician defined state is not correctly identified as seen in Fig.S4(b). It is clear from Fig.S4(b) that utilizing the metric $\delta_{\text{DIS}}$ with $\delta_b = 1$ sufficiently detects the number of patients in the ICU admission state without a significant number of false positives.

Note that although Algorithm 1 performs fine-grained personalization for patients, we do not utilize any demographic information (e.g. age, sex, comorbidity) of the patient as it is not available in the current dataset. In future work we will combine our fine-grained personalization Algorithm 1, then utilize state-of-the-art personalization techniques in (S1) to refine the clinical state estimates of the patient's by only comparing patients with similar demographic information.

S4(a) Dynamic models that do not contain final clinical state at discharge or ICU admission.

S4(b) Dynamic models that contain the final clinical state at discharge or ICU admission.

Figure S4: Compute $\delta_{\text{ICU}}$ and threshold $\delta_b = 1$ for Algorithm 1 for estimating the clinical state of the 1065 patients.

## S5    Complexity and Implementation of Algorithm 1 in Hospital Wards

The two most computationally expensive operations in Algorithm 1 are the non-parametric Bayesian inference and the similarity comparison for large sets of electronic health record data. In this section we provide methods to efficiently implement Algorithm 1 for large datasets.

To address the complexity of the non-parametric Bayesian inference several state-of-the-art sampling methods can be utilized. The Gibbs sampler [7, 33] has a computational complexity off $O(|T^i||\mathcal{K}^i|)$ with $|T^i|$ the number of samples for patient $i$, and $|\mathcal{K}^i|$ the number of unique dynamic models associated with patient $i \in \mathcal{I}$. Typical for medical data the number of samples $|T^i|$ and associated number of clinical states, which is of the same order of magnitude as $|\mathcal{K}^i|$, are sufficiently small allowing the implementation of the Gibbs sampler on standard computing workstations. For example, on a standard desktop computer a patient with $|T^i| = 1000$, $|\mathcal{K}^i| = 10$, and utilizing 10,000 iterations in the Gibbs sampler, the non-parametric Bayesian segmentation completes in under 5 min. There are however several other sampling methods that can be utilized to increase the efficiency of sampling. For example, it may be possible to reduce the number of iterations necessary to sufficiently converge by utilizing the Beam sampler technique introduced in [33]. The Beam sampler is constructed by combining the ideas of slice sampling and dynamic programming to sample the

whole hidden state trajectories $\{z_t\}_{t \in T}$, however the Beam sampler has computational complexity $O(|T^i||\mathcal{K}^i|^2)$. For online estimation, streaming methods for dynamic model estimation are desirable. Streaming variational inference, which utilizes assumed density filtering, has been introduced in [27] to dynamically update the dynamic model parameter estimates as new samples arrive. In [3] an online variational technique is introduced, based on a split-merge topic update routine, for updating the parameter estimates as new data arrives. To account for asynchronous data arrival, which is typical in medical settings, a unique posterior decomposition method in a combinatorial optimization framework can be used to update the posterior distribution as new data from each stream is received [4]. The selection of which sampling method to utilize depends on the application setting. For our analysis we utilized the Gibbs sampler for all estimates as we are interested in segmenting the dataset $\mathcal{D}$ into $\bar{\mathcal{D}}$ which is performed in the offline stage of Algorithm 1. Additionally for new patients, the number of samples is sufficiently small to allow the Gibbs sampler to be utilized for constructing the dynamic model of the patient which takes less than 5 min on a standard desktop computer. For the results presented in this paper the hyper-parameters of the combinatorial stochastic model are given by: $\gamma = 0.01$, $\alpha = 1.01$, $L = 15$, $\boldsymbol{\mu}_0 = \mathbf{0} \in \mathbb{R}^m$, $\lambda = 1$, and $v = m$.

The computational complexity of evaluating the similarity (i.e. clinical state) of a new patient is given by $O(|\mathcal{L}|m^3)$ where $O(m^3)$ is the computational complexity of evaluating the Bhattacharyya distance. Notice that in medical applications the number of physiological data streams $m$ (e.g. $\boldsymbol{y} \in \mathbb{R}^m$) is typically sufficiently small such that $|\mathcal{L}|$ is the major contributor to the decease in computational efficiency of Algorithm 1. To address this issue the cardinality of $\mathcal{L}$ for large medical datasets, that may be composed of millions of patients, can be reduced by introducing a uniqueness threshold $\delta_u$. If a new segment $k$ does not satisfy

$$\min_{k' \in \mathcal{L}} \{D_B(k, k')\} > \delta_u \tag{S2}$$

then it is not added to the dataset $\mathcal{L}$ as it is not sufficiently unique from the segments already contained in $\mathcal{L}$. It is expected that several patients in a large medical dataset will share similar physiological signals that are associated with the same clinical state.

The computational complexity of Algorithm 1 is polynomial with $O(|\mathcal{I}| \max(|T^i|) \max(|\mathcal{K}^i|))$ the computational complexity of the segmentation of the EHR data to construct the labeled dataset $\mathcal{L}$, and $O(|\mathcal{L}|m^3)$ the computational complexity of computing the clinical state of a new patient.

## S6   Proof of Theorem 1

The proof of Theorem 1 is based on using the matrix Bernstein inequality presented in [32]. The main idea is to use a linear mapping that allows the construction of a vector Bernstein inequality from the matrix Bernstein inequality. Here we define the linear mapping $L : \mathbb{R}^m \to \mathbb{R}^{m \times m}$ by

$$L(Y) = \begin{bmatrix} 0 & Y' \\ Y & 0 \end{bmatrix}. \tag{S3}$$

Set $X_t = L(Y_t)$ and note that $X_t$ are real and symmetric random matrices with

$$X_t^2 = \begin{bmatrix} ||Y_t||_2^2 & 0 \\ 0 & Y_t Y_t' \end{bmatrix}. \tag{S4}$$

Since, $||L(Y)|| = ||Y||_2$ we have that $||X_t|| = ||Y_t||_2 \leq L$. Additionally,

$$\left\| \sum_{t=1}^n E[X_t^2] \right\| = \begin{bmatrix} \sum_{t=1}^n E[||Y_t||_2^2] & 0 \\ 0 & \sum_{t=1}^n E[Y_t Y_t'] \end{bmatrix} = \sum_{t=1}^n E[||Y_t||_2^2]. \tag{S5}$$

Plugging these relations into the matrix concentration inequality in [32], Theorem 1 results.   □

The corollary of Theorem 1 for real and symmetric matrices is given by:

**Corollary 1** *Let $\{Y_1, \ldots, Y_n\}$ be a set of independent random real and symmetric matrices with $Y_t \in \mathbb{R}^{m \times m}$ for $t \in \{1, \ldots, n\}$. Assume that each has uniform bounded deviation such that $||Y_t|| \leq L \quad \forall t \in \{1, \ldots, n\}$. Writing $Z = \sum_{t=1}^n Y_t$, then*

$$P(||Z|| \geq \varepsilon) \leq (2m) \exp\left(\frac{-3\varepsilon^2}{6V(Z) + 2L\varepsilon}\right), \quad V(Z) = \left\| \sum_{t=1}^n E[Y_t^2] \right\|. \tag{S6}$$

## Supporting Material Reference

[1] E. Alam, N .and Hobbelink, A. van Tienhoven, P. van de Ven, E. Jansma, and P. Nanayakkara. The impact of the use of the early warning score (EWS) on patient outcomes: a systematic review. *Resuscitation*, 85(5):587–594, 2014.

[2] M. Beal, Z. Ghahramani, and C. Rasmussen. The infinite hidden Markov model. In *Advances in neural information processing systems*, pages 577–584, 2001.

[3] M. Bryant and E. Sudderth. Truly nonparametric online variational inference for hierarchical Dirichlet processes. In *Advances in Neural Information Processing Systems*, pages 2699–2707, 2012.

[4] T. Campbell, J. Straub, J. Fisher, and J. How. Streaming, distributed variational inference for Bayesian nonparametrics. In *Advances in Neural Information Processing Systems*, pages 280–288, 2015.

[5] L. Clifton, D. Clifton, P. Watkinson, and L. Tarassenko. Identification of patient deterioration in vital-sign data using one-class support vector machines. In *2011 Federated Conference on Computer Science and Information Systems*, pages 125–131. IEEE, 2011.

[6] D. Commenges, P. Joly, L. Letenneur, and J. Dartigues. Incidence and mortality of alzheimer's disease or dementia using an illness-death model. *Statistics in medicine*, 23(2):199–210, 2004.

[7] E. Fox, E. Sudderth, M. Jordan, and A. Willsky. An HDP-HMM for systems with state persistence. In *Proceedings of the 25th international conference on Machine learning*, pages 312–319. ACM, 2008.

[8] R. Gentleman, J. Lawless, J. Lindsey, and P. Yan. Multi-state Markov models for analysing incomplete disease history data with illustrations for HIV disease. *Statistics in medicine*, 13(8):805–821, 1994.

[9] S. Ghosal and A. Van der Vaart. Fundamentals of nonparametric Bayesian inference, 2015.

[10] J. Ghosh and V. Ramamoorthi. Bayesian nonparametrics, 2003.

[11] L. Hjort, C. Holmes, P. Müller, and S. Walker. *Bayesian nonparametrics*, volume 28. Cambridge University Press, 2010.

[12] N. Hjort. Nonparametric Bayes estimators based on beta processes in models for life history data. *The Annals of Statistics*, pages 1259–1294, 1990.

[13] H. Ishwaran and L. James. Gibbs sampling methods for stick-breaking priors. *Journal of the American Statistical Association*, 2011.

[14] C. Jackson and L. Sharples. Hidden Markov models for the onset and progression of bronchiolitis obliterans syndrome in lung transplant recipients. *Statistics in medicine*, 21(1):113–128, 2002.

[15] C. Jackson, L. Sharples, S. Thompson, S. Duffy, and E. Couto. Multistate Markov models for disease progression with classification error. *Journal of the Royal Statistical Society: Series D (The Statistician)*, 52(2):193–209, 2003.

[16] R. Kay. A Markov model for analysing cancer markers and disease states in survival studies. *Biometrics*, pages 855–865, 1986.

[17] W. Knaus, E. Draper, D. Wagner, and J. Zimmerman. APACHE II: a severity of disease classification system. *Critical care medicine*, 13(10):818–829, 1985.

[18] X. Nguyen. Borrowing strength in hierarchical Bayes: Posterior concentration of the Dirichlet base measure. *Bernoulli*, 22(3):1535–1571, 2016.

[19] R. Pérez-Ocón, J. Ruiz-Castro, and L. Gámiz-Pérez. Non-homogeneous Markov models in the analysis of survival after breast cancer. *Journal of the Royal Statistical Society: Series C (Applied Statistics)*, 50(1):111–124, 2001.

[20] M. Pimentel, D. Clifton, L. Clifton, and L. Tarassenko. A review of novelty detection. *Signal Processing*, 99:215–249, 2014.

[21] J. Quinn and C. Williams. Known unknowns: Novelty detection in condition monitoring. In *Pattern Recognition and Image Analysis*, pages 1–6. Springer, 2007.

[22] M. Rothman, S. Rothman, and J. Beals. Development and validation of a continuous measure of patient condition using the electronic medical record. *Journal of biomedical informatics*, 46(5):837–848, 2013.

[23] S. Saria, D. Koller, and A. Penn. Learning individual and population level traits from clinical temporal data. In *Proc. Neural Information Processing Systems (NIPS), Predictive Models in Personalized Medicine workshop*. Citeseer, 2010.

[24] E. Solberg and A. Lahti. Detection of outliers in reference distributions: performance of horn's algorithm. *Clinical chemistry*, 51(12):2326–2332, 2005.

[25] P. Subbe, M. Kruger, P. Rutherford, and L. Gemmel. Validation of a modified Early Warning Score in medical admissions. *Qjm*, 94(10):521–526, 2001.

[26] M. Sweeting, V. Farewell, and D. De Angelis. Multi-state Markov models for disease progression in the presence of informative examination times: An application to hepatitis C. *Statistics in medicine*, 29(11):1161–1174, 2010.

[27] A. Tank, N. Foti, and E. Fox. Streaming variational inference for Bayesian nonparametric mixture models. 2015.

[28] L. Tarassenko, A. Hann, and D. Young. Integrated monitoring and analysis for early warning of patient deterioration. *British journal of anaesthesia*, 97(1):64–68, 2006.

[29] Y. W. Teh. Dirichlet process. In *Encyclopedia of machine learning*, pages 280–287. Springer, 2011.

[30] Y. W. Teh, M. Jordan, M. Beal, and D. Blei. Hierarchical Dirichlet processes. *Journal of the american statistical association*, 2012.

[31] R. Thibaux and M. Jordan. Hierarchical beta processes and the indian buffet process. In *International conference on artificial intelligence and statistics*, pages 564–571, 2007.

[32] J. Tropp. An introduction to matrix concentration inequalities. *Foundations and Trends in Machine Learning*, 8(1-2):1–230, 2015.

[33] J. Van Gael, Y. Saatci, Y. W. Teh, and Z. Ghahramani. Beam sampling for the infinite hidden Markov model. In *Proceedings of the 25th international conference on Machine learning*, pages 1088–1095. ACM, 2008.