[Reviews · NeurIPS 2016]

Reviewer 1

Summary

NA

Qualitative Assessment

Thank you for the submission. There are a few points of clinical clarity. (1) The use of the term clinical state lacks specificity in the scope of this paper. The more targeted goal that is in line with other models in this area is to predict clinical deterioration (a shift in patient condition that requires a higher level of care). The goal of "clinical state" is less about understanding what it is but how shifts in the state drive an intervention (i.e. ICU transfer) (2) The authors should clarify the use case they describe. Though care practices may vary, in general, patients of these types receive treatment on an outpatient basis (i.e. they would come to the hospital daily to receive their treatment and then go home). Please confirm with your data providers whether or not this is the case. If so, I would modify the x axis of Figure 2a and 2b to be days (rather than hours). Hours seems to imply real time telemetry like data or an inpatient stay where vitals are checked routinely throughout the day and night. The authors do refer to admission time from 4 days to 85 days in some cases, but I'm not sure why these patients were admitted to begin with. It seems that there is some other inciting event that put them in the hospital (aside from just getting chemotherapy treatment). (3) If the authors confirm that the data is outpatient data, then a note should be made that the comparison to MEWS and Rothman index are somewhat apples to oranges in that both are designed for inpatient settings. However the comparison is fine as they are both aggregate scores of clinical conditions. (4) The authors hint at the models doing prediction with data 12 hours before an ICU admission or discharge. Is this in is a cross-validation framework? It seems like the same dataset is used to train and test the models so it is unclear what the model is missing. (5) Did the patient actually go to the ICU or are these estimates for what the labeling clinician thought as a set of vitals that actually required an ICU admission (i.e. the physicians were not given any clinical details and simply given the 17 temporal physiologic streams)? (6) line 252, I think you are referring to the wrong figure. (7) I'm a bit confused as to the defined priors. In the context of this study, the outcome is known (ICU vs discharge). In a practical application, there will be long runs when the next 12 or 24 hours will not result in any change. It would seem the prior distribution of the ICU conversion would be much smaller than indicated by the test data.

Confidence in this Review

1-Less confident (might not have understood significant parts)


Reviewer 2

Summary

This paper proposed an algorithm for estimating patient's clinical state based on previously recorded EHR data. The algorithm learns a combinatorial stochastic model for each patient using an a HDP-HMM based non-parametric Bayesian learning method. A data-driven bound on the model complexity is given. The patient's personalized dynamics have been learnt well with the proposed algorithm.

Qualitative Assessment

This paper proposed an algorithm for estimating patient's clinical state based on previously recorded EHR data. The algorithm learns a combinatorial stochastic model for each patient using an a HDP-HMM based non-parametric Bayesian learning method. A data-driven bound on the model complexity is given. The patient's personalized dynamics have been learnt well with the proposed algorithm. The paper is well written and the method is clearly described. The method can be well used in reality and the personalized dynamics of the patients are very important to doctors. The motivation and innovation are clear.

Confidence in this Review

2-Confident (read it all; understood it all reasonably well)


Reviewer 3

Summary

This paper provides a non-parametric learning algorithm for discovering the dynamic model of patients and classifying their associated clinical states.

Qualitative Assessment

This paper provides a non-parametric learning algorithm to estimate patients' clinical states. Although the soundness of the techniques seems to be fine, the novelty of the approach may not be enough for NIPS standard since the work employs mostly off-the-shelf techniques to form a system instead of devising new algorithms. The most attractive part of this paper is that the techniques may be used in practical. And the experiments are quite solid.

Confidence in this Review

2-Confident (read it all; understood it all reasonably well)


Reviewer 4

Summary

To my best understanding, this paper applies the HDP-HMM model of Fox et al to clinical data, with the aim of better modelling and hence better prediction of ICU admission. The main contribution of this paper is Algorithm 1, which is personalized for each patient. Additionally, this paper presents several statistical tests for model validation, which is rare in machine learning papers, it is good to see these here. The experiments show superior performance of the proposed Algorithm 1, against several baselines, some of which are state-of-the-art.

Qualitative Assessment

This paper is rich in content. However, it is also very technical and thus I am not able to provide in depth review to the theoretical results. One of my major concern with this work is that various statistical tests are proposed, but I cannot find any experimental results on the tests (in both the paper and the supporting material). This suggests that a significant portion of the experiments is missing. In section 2, it will be informative to include an example of patient vitals y_t. Some examples from the supporting material can be replicated here. Additionally, it would be nice to provide more details on the cancer ward dataset, perhaps by listing all the vital signs in a table and provide a statistic of their values (range, mean, median, etc). I believe this is an excellent paper and I vote for acceptance once the authors address my concerns above. Some nitpicks below: 1. The acronym of switching multivariate Gaussian is given as SGM rather than SMG. Why is this so? Is it because SGM sounds better? 2. Referencing the supporting material 'section' will make finding the relevant material easier. (For example, "provided in the Supporting Material S#") 3. Proper nouns in the reference list are not properly capitalized. Examples include "Dirichlet", "Bayesian", "Bernstein", "Markov". Teh's name in the reference list is not correct, it should be Y. W. Teh. Also Ghahramani is missing first initial Z. 4. Inline citations should not be used as nouns, including the authors before the citation will make reading easier. For example, in page 1, "methods in [17]" -> "methods in Pimentel [17]".

Confidence in this Review

1-Less confident (might not have understood significant parts)


Reviewer 5

Summary

This paper presents a non-parametric learning algorithm to estimate the clinical state of a patient.

Qualitative Assessment

This paper looks promising with potential applications. Though I am not an expert in this specific area, I would like to draw one important observation, i.e. regarding inertial HMM. Has the author(s) taken into account the criticisms of Fox et al regarding the persistence of state? Please look at this paper: http://www.aaai.org/ocs/index.php/AAAI/AAAI15/paper/view/9475/9470 - can you incorporate this to your model? What is the computational complexity as compared to those methods stated in Table 1 ?

Confidence in this Review

2-Confident (read it all; understood it all reasonably well)


Reviewer 6

Summary

A system combining the merits of non-parametric Bayesian inference, permutation testing, and generalizations of the empirical Bernstein inequality is proposed to model patient's clinical state and dynamics. The Hierarchical Dirichlet Process Hidden Markov Model (HDP-HMM) is used to model the dynamics. The generalization of the Bernstein inequality is used to give the model parameters statistical guarantees.

Qualitative Assessment

It looks like an application-driven paper. The key modeling part about patent's clinical state dynamics is based on HDP-HMM. The key modeling part and the permutation testing part are independent of each other and are just two components of a system for solving the problem.

Confidence in this Review

1-Less confident (might not have understood significant parts)